# Structures of the eukaryotic ribosome and its translational states in situ

Patrick C. Hoffmann[1,4], Jan Philipp Kreysing [1,2,4], Iskander Khusainov [1,4], Maarten W. Tuijtel [1], Sonja Welsch [3] & Martin Beck [1] ✉

Ribosomes translate genetic information into primary structure. During translation, various cofactors transiently bind to the ribosome that undergoes prominent conformational and structural changes. Different translational states of ribosomes have been well characterized in vitro. However, to which extent the known translational states are representative of the native situation inside cells has thus far only been addressed in prokaryotes. Here, we apply cryo-electron tomography to cryo-FIB milled *Dictyostelium discoideum* cells combined with subtomogram averaging and classification. We obtain an in situ structure that is locally resolved up to 3 Angstrom, the distribution of eukaryotic ribosome translational states, and unique arrangement of rRNA expansion segments. Our work demonstrates the use of in situ structural biology techniques for identifying distinct ribosome states within the cellular environment.

Protein translation by ribosomes is a key mechanism across all living organisms. The translation cycle can be divided into four main steps: initiation, elongation, termination and recycling. Incorporation of every new amino acid into the nascent chain during elongation requires multiple conformational rearrangements of the ribosome and changes to interactions with multiple proteins (translation factors) and RNAs. The factors of the translation cycle involved in elongation are conserved between bacteria and eukaryotes[1].

Numerous structures of ribosomes solved by crystallography and cryo-electron microscopy (cryo-EM) have provided a detailed view on rearrangements and their interactions with protein and RNA cofactors during translation (reviewed in[2]). In these studies, the functional state of interest is specifically enriched for structural analysis, often from bacterial species. By combining cryo-electron tomography (cryo-ET) with subtomogram averaging (STA), the translational states inside bacterial *Mycoplasma pneumoniae* cells were elucidated at sub-nanometer resolution[3]. However, *M. pneumoniae* cells are exceptionally thin and thus can be analyzed directly by cryo-ET subsequent to plunge freezing. Due to sample thickness limitations, larger eukaryotic cells remained inaccessible to this approach. To make such cells accessible to cryo-ET for STA, specimen thinning techniques are used that remove excess material, such as vitreous sectioning or cryo-focused ion beam (cryo-FIB) milling[4,5]. Due to challenges in the experimental and data processing workflow, translational states of eukaryotes have thus far only been determined ex vivo; from purified human polysomes analyzed by single particle cryo-EM[6].

Here, we investigate the eukaryotic ribosome of the model organism *Dictyostelium discoideum*, a member of amoebozoa, one of the major taxonomic groups of eukaryotes, directly by cryo-ET and STA from cryo-FIB milled cells. We determined the in situ structure of the *D. discoideum* 80 S ribosome revealing the specific organization of its long rRNA expansion segments. Using 3D classification, we resolved distinct ribosome states bound to translation factors and tRNAs in the cellular environment and assigned them to the translation cycle.

## Results

In order to visualize the translation machinery directly in *D. discoideum* cells, we prepared lamellae with a thickness of 130–200 nm by cryo-FIB milling of vitrified cells (Fig. 1a). We used lamellae overview images to identify areas suited for cryo-ET acquisition (Fig. 1b, c). For structural

[1]Department of Molecular Sociology, Max Planck Institute of Biophysics, Max-von-Laue-Straße 3, 60438 Frankfurt am Main, Germany. [2]Department of Molecular Sociology, IMPRS on Cellular Biophysics, Max-von-Laue-Straße 3, 60438 Frankfurt am Main, Germany. [3]Central Electron Microscopy Facility, Max Planck Institute of Biophysics, Max-von-Laue-Straße 3, 60438 Frankfurt am Main, Germany. [4]These authors contributed equally: Patrick C. Hoffmann, Jan Philipp Kreysing, Iskander Khusainov. ✉e-mail: martin.beck@biophys.mpg.de

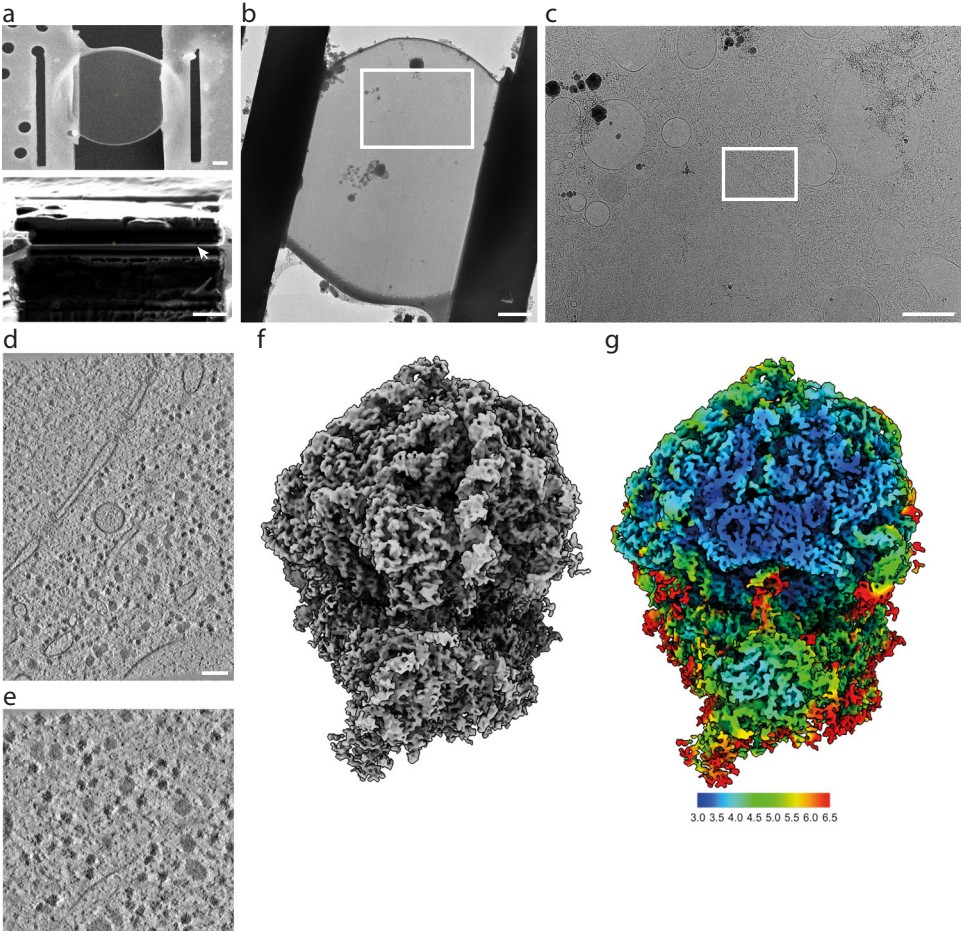

**Fig. 1 | In situ ribosome structure of *D. discoideum*. a** Cryo-FIB milled lamella SEM view (top) and FIB view (bottom) of a *D. discoideum* cell. The lamella front in the FIB view is indicated by the white arrow. **b** Cryo-TEM overview of the same lamella, rotated by -180° compared to (**a** top view), white box indicates the area shown in **c**. **c** Higher magnification cryo-TEM overview showing the subcellular context, white box indicates the area of tilt series acquisition. **d** Virtual slice of one out of 125 tomograms (dataset 1) and from the area shown in **c**, rotated clockwise by 90°. **e** Virtual slice of one out of 98 tomograms (dataset 2) at higher magnification. **f** Subtomogram average of the 80S *D. discoideum* ribosome obtained from dataset 2. **g** Corresponding local resolution map. Scale bars: 2 μm in **a** and **b**, 1 μm in **c**, and 100 nm in **d** and **e**.

analysis of the *D. discoideum* ribosome, we used 125 tomograms from the perinuclear region of 42 cells (dataset 1, 2.2 Å pixel size) (Fig. 1d). Ribosomes in cytosolic areas were identified by template matching and extracted from the tomograms. Using Warp/M-refinement and classification in Relion, we solved the cytosolic 80S ribosome of *D. discoideum* by STA to 4.5 Å resolution from 29858 particles (Supplementary Fig. 1a, b, Supplementary Fig. 2). Because our reconstruction was likely limited by the pixel size, we collected a second dataset of tilt series at higher magnification. We used 98 tomograms of 8 cells (dataset 2, 1.2 Å pixel size) (Fig. 1e). Using the same processing approach, we solved the cytosolic 80S ribosome of *D. discoideum* to 3.8 Å resolution from 24399 particles of dataset 2 (Fig. 1f, g, Supplementary Fig. 1c, d) which was further used for structure interpretation. The 80S ribosome subtomogram average resolves characteristic features of the unrotated 80S with tRNA in P/P position, and the additional density around the A-site belonging to the eEF1A*A/T-tRNA complex and eEF2 (Fig. 2a). In core regions of the 60S subunit, which show the highest local resolution, protein side chains, and individual nucleotides are clearly resolved, and examplarily shown for uL18a and 26S rRNA (Fig. 2b). The structure shows the presence of several rRNA expansion segments on both small and large ribosomal subunits (Fig. 2c). However, since no complete 3D models or 2D diagrams were available, we used density-based homology rRNA modeling to build the complete 2D diagrams of the 17S, 5S, 5.8S and 26S rRNA, and to

resolve the 3D architecture of the rRNA expansion segments on the ribosome. The resulting model allowed us to build the full map of rRNAs of the small (Supplementary Fig. 3) and the large subunit (Supplementary Fig. 4).

The *D. discoideum* cytosolic 80S ribosome combines characteristic expansion segment (ES) arrangements of other representative eukaryotic model organisms such as *Saccharomyces cerevisiae, Drosophila melanogaster, Mus musculus, Sus scrofa*, and *Canis lupus* (Fig. 2c–e, Supplementary Fig. 5a–h)[7–11] in a unique manner. The long 17S rRNA ES9S of *D. discoideum* is located at the head of the small subunit and oriented towards the central protuberance of the large subunit where it approaches the long expansion segment ES7L(D) (Fig. 2c, Supplementary Fig. 5a) of the large subunit. Despite the presence of long ES9S in *D. melanogaster* ribosome[8], and similar three-helical structure of ES7L in *M. musculus*[9], the intersubunit contact formed by these ES is specific for *D. discoideum* (Supplementary Fig. 5a, c, d). The characteristic 26S rRNA expansion ES27L of *D. discoideum* forms a single helix, and reaches the peptide exit tunnel (Fig. 2c, Supplementary Fig. 5a).

In addition to the cytosolic ribosome, we identified by classification a fraction of ribosomes in dataset 1 with density for associated cellular membrane (3314 particles, Supplementary Fig. 2, Supplementary Fig. 5e). Due to the preferential acquisition of the perinuclear region in dataset 1, a larger fraction of ribosomes was directly bound to

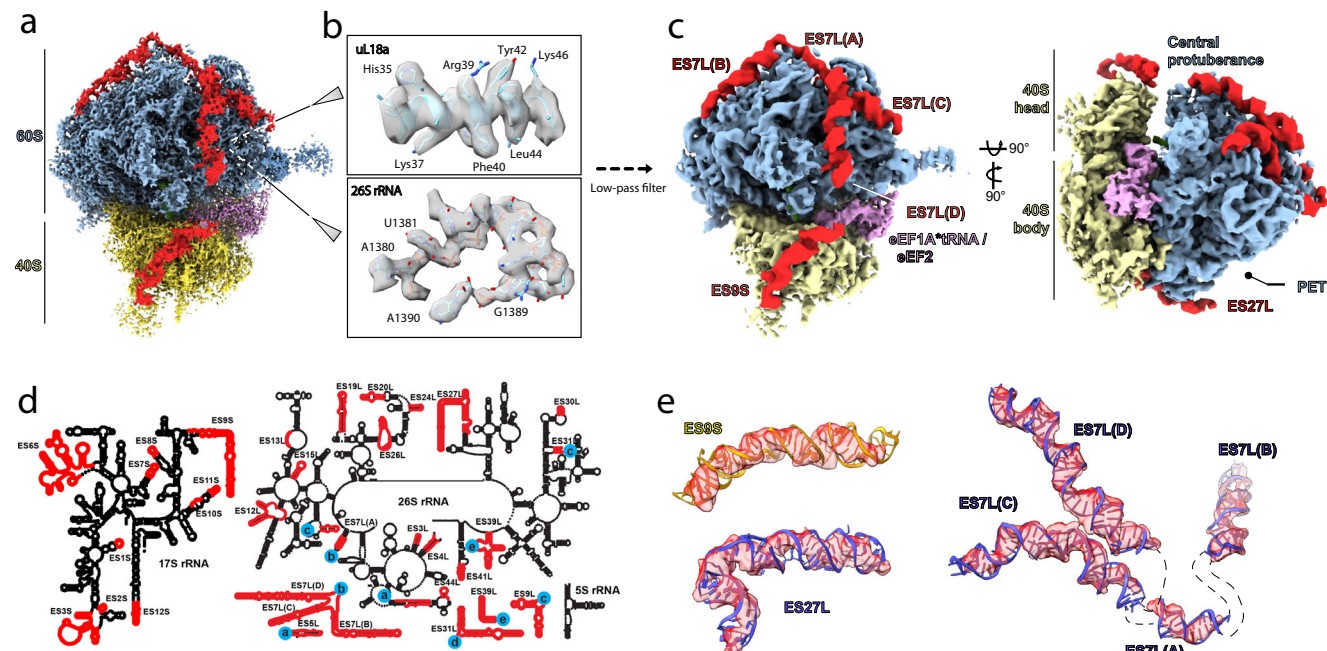

**Fig. 2 | Visualization of the expansion segments on the 80S ribosome. a** 3.8 Å structure of 80S ribosome subtomogram average. Expansion segments are colored in red, the density corresponding to the A/T-tRNA, eEF1A, and eEF2 in lavender. **b** Close-up view of the density in the 60S core of protein uL18a and a fragment of 26S rRNA with fitted models. **c** Gaussian-filtered (sDev = 2) 80S ribosome density map showing the close proximity between ES9S and ES7L(D), and the protrusion of ES27L towards the peptide exit tunnel (PET). **d** Schematic representation of the 2D diagrams of rRNAs. Expansion segments are colored in red. **e** Accurate fit of the rRNA expansion segments into the Gaussian-filtered (sDev = 2) cryo-ET density segments. Unmodeled residues are represented by dashed lines. Color legend: Blue, 60S subunit. Yellow, 40S subunit. Red, expansion segments. Lavender, eEF1A and eEF2.

the nuclear envelope rather than to the endoplasmic reticulum. 3D classification of these particles could not resolve structural heterogeneity. The membrane-bound ribosome consensus map showed a luminal density proximal to the membrane. Comparison to a mammalian membrane-bound ribosome in complex with the translocon Sec61, the translocon-associated protein (TRAP) complex and oligosaccharyl transferase (OST) complex bound (EMD-3069[11]), suggests that this luminal density might belong to the TRAP complex (Supplementary Fig. 5e, f). However, in the membrane-bound *D. discoideum* structure we do not observe density for OST or for ES27L, which in its original position on the large subunit (Fig. 2c, Supplementary Fig. 5a) would sterically clash with the membrane (Supplementary Fig. 5a, e–h).

Further, we performed 3D classification of the large fraction of cytosolic ribosomes (29858 particles) from 42 different cells from dataset 1 to investigate the structural heterogeneity and to identify individual translational states. This approach revealed ten distinct ribosome structures. With the help of PDB models of ribosomes with associated factors from different species[12–16], we could assign seven of them to states of the translation-elongation cycle and translation initiation (Fig. 3a, Supplementary Fig. 2, Supplementary Fig. 6a–g, Supplementary Fig. 7). The largest fraction of 80S ribosomes belonged to steps preceding peptide bond formation (Fig. 3a, Supplementary Fig. 6a, b), where we could resolve the stages of binding of aminoacyl eEF1A*tRNA ternary complex to the ribosome, A-tRNA recognition and its proofreading by the 80S ribosome. The second largest fraction corresponded to the peptidyl transfer step (Fig. 3a, Supplementary Fig. 6f). We identified three states of tRNA translocation facilitated by ribosome ratcheting and elongation factor eEF2. Two of these states show different conformations of eEF2 and A/P- and P/E-tRNAs (Fig. 3a, Supplementary Fig. 6c, d) comparable to prokaryotic translational states with EF-G[3,17] and one state has eEF2 and P and E-tRNA bound (Fig. 3a, Supplementary Fig. 6e). The dataset also contained a small

fraction of the initiation 80S ribosome (Fig. 3a, Supplementary Fig. 6g). In addition, we observed three factor-bound states, which we could not directly place in the translation cycle (Fig. 3b). This included two eEF2 bound 80S states without tRNA that differed in density at the previously described binding site for eIF5A[18–20] (Fig. 3b, Supplementary Fig. 8a, b). One other state contained A/A and P/P-tRNA, and an additional density at the translation factor binding site, which could not be immediately attributed to eEF1A or eEF2 in its elongated form (Fig. 3b, Supplementary Fig. 8c). About one-tenth of all ribosome particles could not be assigned into a specific class using this workflow.

Lastly, we investigated the variability in translation across different cells to assess robustness of identified translational states across dataset 1, which contained one to eight tomograms per cell from a total of 42 cells. We found that in this dataset translation elongation states are evenly distributed in the perinuclear cytoplasm and comparably populated across different cells (Supplementary Fig. 9a, b). The 80S*eEF2 ribosome substate 1 with density at the eIF5A binding site and the class of unassigned ribosomes showed the largest variation in abundance across cells, indicating some degree of heterogeneity.

## Discussion

Subtomogram averaging approaches of eukaryotic macromolecular complexes in situ have provided valuable insights in native cellular structures[21,22]. However, the attainable resolution is limited by experimental parameters such as sample thickness and signal-to-noise ratio, as well as data acquisition efficiency and particle number. By capitalizing on current advances of in situ cryo-ET sample preparation and image processing, we were able to solve the structure of the eukaryotic 80S ribosome up to side-chain resolution at its core from inside of *D. discoideum* cells. The respective dataset was collected within a single 72-hour cryo-ET session. These results emphasize the potential of in situ structural biology to resolve macromolecular complexes directly inside of eukaryotic cells at high resolution.

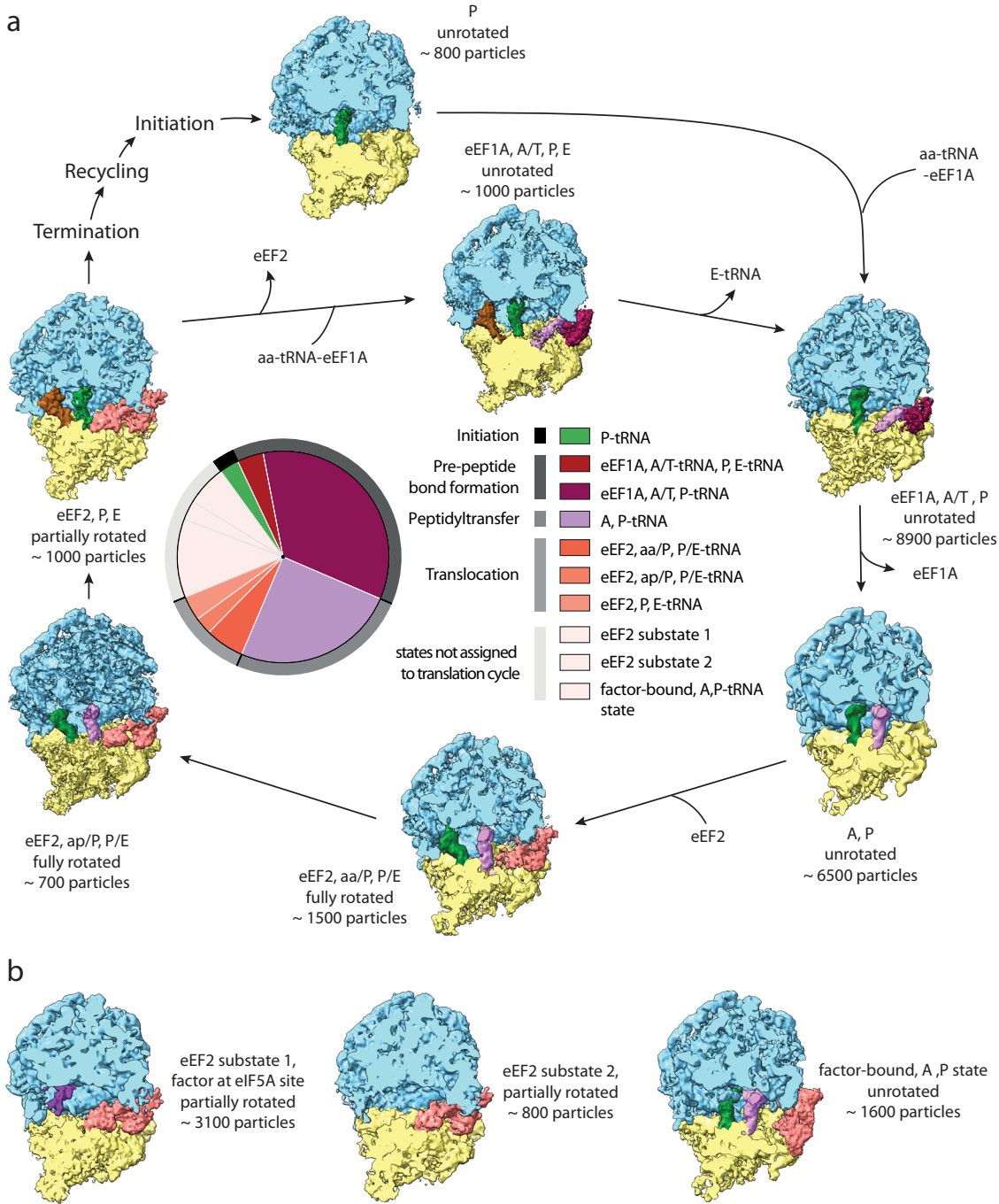

**Fig. 3 | Translational states of the 80S eukaryotic ribosome in situ. a** Schematic of the ribosomal translation cycle. Seven ribosome states of the translation cycle could be identified. The two most prominent states are (1) eEF1A, A/T, P and (2) A, P with a third and a quarter of the particles, respectively. Particle numbers for each state are rounded to the next hundred. The tRNA positions are labeled in accordance to the proximity of their anticodon stem-loop (ASL) to the canonical tRNA binding sites on the small subunit. The state aa/P refers to an intermediate state of tRNA for which ASL is closer to the canonical A-site, than the P-site. The ap/P state indicates the progression of the tRNA closer to the P-site. Each state's occurrence is represented in the central pie chart as a fraction of particles calculated from 26649 positively identified particles. Source data are provided as a Source Data file. **b** Three additional 80S ribosome states were identified but could not be placed in the translation cycle and are shown with putative factors. Color legend: Blue, 60S subunit. Yellow, 40S subunit. Lavender, A-tRNA. Green, P-tRNA. Brown, E-tRNA. Maroon, eEF1A. Salmon, eEF2 or undetermined factor at the translation factor binding site in **b**, right panel. Purple, putative factor at eIF5A site.

We visualized peripheral regions such as rRNA expansion segments (ES) and the P-stalk (Fig. 2c). Notably, ES are variable in sequence and size across eukaryotes and their functions remain poorly understood. Recent studies showed that ES might be required for proper cell growth and participate in ribosome biogenesis, translation fidelity, protein folding, ER membrane binding and interactions with the mRNA and auxiliary proteins (reviewed in[23]). However, most of the studies were performed using *S. cerevisiae* as a model, which has shorter ES compared to higher eukaryotes[1]. Therefore, we compared the structure of the *D. discoideum* 80S ribosome to various structures of other eukaryotic model organisms, e.g., *S. cerevisiae, Drosophila melanogaster, Mus musculus, Sus scrofa* and *Canis lupus*[7-11] (Supplementary Fig. 5a–h). We specifically focused our analysis on three ES – ES9S, ES7L, and ES27L –which are most variable in length and

conformation between species. In contrast to selected cryo-EM structures from other organisms, we observe strong and complete density for all three ES. The most striking example is ES9S. ES9S was shown to be involved in regulation of gene expression in mouse neuronal cells via sequence-dependent direct contact between rRNA and *Hoxa9* mRNA[24]. In *D. discoideum*, ES9S is longer and may play a role in stabilizing the ribosome by contacting ES7L(D) of the large subunit (Fig. 2c, Supplementary Fig. 5a). Of note, ES7L(D) is not always well resolved in the structures of other species despite being present in the sequence, possibly due to its flexibility (Supplementary Fig. 5b, c). Lastly, the conformation of the ES27L may depend on ribosome localization within the cell. For the *D. discoideum* 80S cytosolic ribosome (Fig. 2c, Supplementary Fig. 5a), ES27L is protruding towards the exit tunnel of the ribosome, where it could support translation fidelity by recruiting nascent peptide processing enzymes, as was shown for *S. cerevisiae* (Supplementary Fig. 5h)[25,26]. In the membrane-bound fraction of the *D. discoideum* 80S ribosome (Supplementary Fig. 5e), ES27L was only partly resolved, indicating its displacement due to steric overlap with the membrane. Although ES27L was shown to facilitate attachment of the ribosome to microsomes derived from dog pancreas[27], we do not observe any density in the corresponding region for *D. discoideum* (Supplementary Fig. 5e), nor in the other selected ribosomal complexes with Ebp1, or Sec61 (Supplementary Fig. 5d, g). Interestingly, ES27L in the 80S-OST-TRAP complex from *C. lupus*[11] is in a different conformation compared to *D. discoideum* or *Sus scrofa* (Supplementary Fig. 5a, e–g). Based on these data we suggest that ES27L is flexible and may adopt various conformations depending on interaction context and cellular localization.

The rRNA ES are considered to play a regulatory role in translation, and their proportion progressively increases from lower to higher eukaryotes[23]. In addition, the natural habitat of the organism is likely to influence the amount of ribosomal regulatory elements, especially expansion segments[28]. Together with other recent studies[29–31], our data support this notion, showing that slime mold contains several long ES characteristic to higher eukaryotes, whereas eukaryotic parasites have a significantly reduced amount of rRNA ES.

For the membrane-bound *D. discoideum* 80S ribosome we observe a density at the luminal side of the membrane, probably attributed to TRAP complex (Supplementary Fig. 5e), but not the glycosylation complex OST, which has been observed in the mammalian 80S-TRAP-OST complex (Supplementary Fig. 5f)[11]. For mammalian OST, the different catalytic subunits STT3A and STT3B were described to glycosylate either co-translationally on the nascent polypeptide or post-translationally[32]. In *D. discoideum* only one form of the OST subunit STT3 is conserved. The absence of luminal OST in our structure suggests that post-translational N-glycosylation may be the dominant mode for proteins inserted into the nuclear envelope and endoplasmic reticulum in *D. discoideum*.

The most populated translation-elongation states of the cytosolic *D. discoideum* ribosome were the pre-peptide bond formation and the peptidyl-transfer steps, demonstrating coherence of the in situ cryo-ET results with kinetic studies[33] and in silico simulations[34], even though most of those were obtained using bacterial systems. A recent study of in situ translation in prokaryotic *Mycoplasma* cells observed the largest fractions of ribosomes in the peptidyl-transfer step and the EF-Tu bound pre-peptide bond formation steps[3]. Thus, prokaryotic *Mycoplasma* cells and eukaryotic *D. discoideum* cells show notable similarity in their in vivo distribution of translation-elongation states. Despite that, possible differences of translation states between other species should be considered.

In comparison, quantification of translational states of human ex vivo purified polysomes using SPA[6] has resolved seven elongation and one pre-recycling state of the 80S. However, only one class (~8% of total particles) contained elongation factor eEF1A and the elongation factor eEF2 was not present[6]. In contrast, in our in situ *D. discoideum*

dataset, the majority of ribosome states have elongation factors bound. Using the in situ approach, we find more than a third of ribosomes engaged at the pre-peptide bond formation steps, states which could not be resolved by the ex vivo approach. We could not observe all states that had been identified ex vivo, possibly due to low abundance of the respective particles. Therefore, both ex vivo and in situ studies have their benefits and result in a complementary view of the conformational landscape of ribosomes.

Besides the well-described states of the eukaryotic translation-elongation cycle, we identify a state with a compact density at the translation factor binding site and A/A and P/P-tRNA. Due to lack of atomic models and the limited local resolution of the factor density, it is not possible to confidently determine its identity. However, fitting of the individual domains of eEF2 into the density may suggest a compact conformation of eEF2, which had been described to exist for the bacterial factor EF-G[35,36].

The 80S*eEF2 states lacking any tRNA ligands are unlikely to participate in translation-elongation. Based on the fit of the atomic model, we assigned the density observed at the factor binding site as eEF2 (Supplementary Fig. 8a, b). However, we cannot exclude that another structurally related factor is bound to this state. For several species, different ribosome structures without tRNA ligands, but bound to factors such as eEF2, Habp4, SERBP1, Stm1, Lso2 and eIF5A alone or in combination have been linked to stabilization of dormant ribosomes[18,37–39]. *D. discoideum* has genes encoding for a member of the Habp4 protein family and for eIF5A. Our 80S*eEF2 substate 1 map indicates that a binding factor is present between the L1 stalk and the binding site of P-site tRNA, which resembles eIF5A in shape (Supplementary Fig. 8b). However, further analysis is needed to clarify the cellular role of the eEF2 substates in *D. discoideum* and if they may progress into functional states of the translation-elongation cycle.

Based on the abundance of cellular ribosomes in tomographic in situ data and potential for cellular heterogeneity of ribosomal states, we envision that the quantification of translational states may become a fingerprint for cellular physiology that will be routinely obtained in the future during the tomographic analysis of cells.

## Methods

### Cell culture

The axenic *D. discoideum* strain Ax2-214 used in this study carried randomly integrated GFP-Nup62 through a pDEX vector with G418/neomycin resistance cassette[40]. Cells were grown in HL5 medium (Formedium) containing 50 μg/mL ampicillin and 20 μg/mL geneticin G418 (Sigma Aldrich) at $20 \pm 2$ °C. Cells were kept either as adherent cells in sub confluent conditions or as suspension culture at a cell density between $1 \times 10^5$ cells/ml to $4 \times 10^6$ cells/ml. Cells were sub-cultured for a maximum of four weeks before re-growing them from cryo-stocks. Before grid preparation, cells were adjusted to a concentration of 2-3 $\times 10^5$ cells/ml and allowed to adhere for 2–4 h prior to cryo-fixation.

### Cryo-EM sample preparation

EM support grids (Au grids 200 mesh, carbon or SiO$_2$ foil, R2/2 or R1/4, all from Quantifoil) were glow discharged with a Pelco easiGlow glow discharger for 90 s at 15 mA. Exponentially growing cells were adjusted to a concentration of $2–3 \times 10^5$ cells/ml. A droplet of 100 μl cell suspension was placed on the glow-discharged grids and cells were allowed to attach to the grid for 2–4 h. Cells were then vitrified by plunge freezing into liquid ethane using a Leica EM GP2 plunger. Lamellae were prepared by cryo-FIB milling with an Aquilos microscope (Thermo Scientific) similar to a previously described workflow[41]. In brief, samples were coated with an organometallic platinum layer using a gas injection system for 10 s and additionally sputter coated with platinum at 1 kV and 10 mA current for 10 s. SEM imaging was performed with 2–10 kV and 13 pA current to check the milling

progress. Milling was performed stepwise with an ion beam of 30 kV while reducing the current from 500 pA to 30 pA. Final polishing was performed with 30 pA current with a lamellae target thickness between 130–200 nm. Some grids were sputter coated with platinum after polishing for 1–2 s at 1 kV and 10 mA.

## Cryo-ET acquisition

The first cryo-ET dataset was acquired in five independent 48-hour microscope sessions from 14 grids from overall 9 independent plunge freezing sessions and with a total of 87 cryo-FIB milled lamellae (Supplementary Table 1). Data were collected at 300 kV on a Titan Krios G2 microscope (Thermo Scientific) equipped with a Gatan BioQuantum-K3 imaging filter in counting mode. For each grid, montaged grid overviews were acquired with 142 nm pixel size. Montages of individual lamellae were taken with 3.9 nm or 2.8 nm pixel size. Tilt series were acquired using SerialEM (version 3.8.1)[42] in low dose mode as -6 K x 4 K movies of 10 frames each, and motion-corrected in SerialEM on-the-fly. The magnification for projection images of 42000x corresponded to a pixel size of 2.176 Å. Tilt series acquisition started from the lamella pretilt of ±8° and a dose symmetric acquisition scheme[43] with 2° increments grouped by 2 was applied, resulting in 59–61 projections per tilt series with a constant exposure time and total dose between 132–150 e$^-$/Å$^2$. The energy slit width was set to 20 eV and the nominal defocus was varied between −2.5 to −5 μm. Dose rate on the detector was targeted to between -10–20 e$^-$/px/s.

The second Cryo-ET dataset was acquired in a single 72-hour session, from 5 lamellae on a single grid (Supplementary Table 1). Data were collected at 300 kV on a Titan Krios G4 microscope equipped with a cold FEG, Selectris X imaging filter, and Falcon 4 direct electron detector, operated in counting mode (all Thermo Scientific). For each grid, montaged grid overviews were acquired with 205 nm pixel size. Montages of individual lamellae were taken with 3.0 nm pixel size. Tilt series projections were acquired using SerialEM (version 4.0.1) in low dose mode as 4K x 4K movies of 10 frames each and on-the-fly motion-corrected in SerialEM. The magnification for projection images of 105,000x corresponded to a pixel size of 1.223 Å. Dose symmetric tilt series acquisition started from the lamella pretilt of +8° and a dose symmetric acquisiton scheme with 2° increments grouped by 2 was applied, resulting in 61 projections per tilt series with a constant exposure time and targeted total dose of ~120 e$^-$/Å$^2$. The energy slit width was set to 10 eV and the nominal defocus was varied between −2.5 to −4.5 μm. Dose rate on the detector was targeted to be ~3–6 e$^-$/px/s.

## Image processing

The motion corrected tilt series were corrected for dose exposure as previously described[44] using a Matlab implementation that was adapted for tomographic tilt series[45]. Poor quality projections were removed after visual inspection. The dose-filtered tilt series were then aligned with patch-tracking in IMOD (versions 4.10.9 and 4.11.5)[46] and reconstructed as back-projected tomograms with SIRT-like filtering of 10 iterations at a binned pixel size of 8.7 Å for dataset 1 and 4.9 Å for dataset 2. From the reconstructed tomograms, 125 (for dataset 1) and 98 (for dataset 2) were selected through visual inspection based on tomogram thickness and visual quality of easily recognizable features such as membrane bilayers or the platinum sputter coat. For compatibility with Relion 3.1[47] and M[48], these 125 tilt series (or 98 for dataset 2) were reprocessed in Warp[49] with the alignment obtained from IMOD.

## Template matching

For dataset 1, a bin8 ribosome average was generated for template matching using Matlab-based scripts from the TOM toolbox[50] from approx. 1000 manually selected ribosomes. For dataset 2, a bin10 ribosome average was generated for template matching using Relion from approx. 1000 manually selected ribosomes. Template matching was performed with the initial ribosome average on bin8 (17.4 Å/px) (dataset 1) and bin10 (12.2 Å/px) (dataset 2) deconvolved tomograms reconstructed in Warp using Dynamo[51] (dataset 1) and STOPGAP (https://github.com/williamnwan/STOPGAP/) (dataset 2). A cytosol mask for each tomogram was created for dataset 1 in IMOD to exclude template-matching hits from the correlation volume in the areas without mature ribosomes, such as vacuoles, nucleus, and outside the lamellae borders. The top 1000 cross-correlation peaks (dataset 1) and top 800 cross-correlation peaks (dataset 2) from the cytosolic area were extracted. The obtained coordinate file was converted to Warp-compatible star file using dynamo2m toolbox (https://github.com/alisterburt/dynamo2m).

## Ribosome classification and refinement

The 125,000 positions (dataset1) and 78,400 (dataset 2) determined through template matching were extracted as subtomograms in Warp[49]. First, bin4 (8.704 Å/px) (dataset 1) and bin6 (7.338 Å/px) (dataset 2) subtomograms were classified in Relion 3.1[47] to filter out non-ribosomal particles such as membranes and other granular structures. This yielded 33172 ribosomal particles (dataset 1) which were then again classified to distinguish between cytosolic (29858 particles) and membrane-bound 80S ribosomes (3314 particles), and 24399 ribosomal particles for dataset 2. The cytosolic ribosomes were then again classified to attempt a separation of lone 60S subunits from full 80S ribosomes, but multiple strategies (global classification with sphere as reference, focused classification on 40S area after refinement on 60S subunit) only yielded an 80S ribosome, and no isolated 60S subunit. We cannot exclude that individual 60S subunits are present and were missed during template matching.

Next, unbinned (2.176 Å/px) and bin1.1 (2.394 Å/px) subtomograms (dataset 1) and bin1.6 (1.957 Å/px) subtomograms (dataset 2) of the cytosolic 80S ribosomes were refined in Relion. The positions (unbinned for dataset 1 and manually upscaled bin1.1 (1.345 Å/px) for dataset 2) were then imported into M (version 1.0.9)[48] to perform multi-particle refinement of the tilt-series and the ribosome. Geometric and CTF parameters were refined in a sequential manner. This resulted in a 4.5 Å (dataset 1) and 3.8 Å (dataset 2) map of the 80S ribosome. New corrected bin1.1 (2.394 Å/px) subtomograms were extracted from M for dataset 1 and refined in Relion again. These M-corrected subtomograms were used as the basis for the following classifications. First, focused classification (10 classes, T = 4, 35 iterations) with a smooth shape mask covering the A-, P-, E-site tRNA positions was performed. The refinements of each class were then subjected to a second round of focused classification (5 classes, T = 5, 35 iterations) with a smooth shape mask covering the factor binding site next to the A-site tRNA position. The resulting classes were all refined, and atomic models of 80S ribosomes with elongation factors and tRNAs bound[12–14,16] were rigid body fitted into the refined maps to identify different states (Supplementary Fig. 6, Supplementary Fig. 7). The particles from different classes were grouped together if the maps looked similar enough. This resulted in the identification of ten distinct ribosome translational states. The well-resolved maps were then segmented in ChimeraX[52] using the Segger function[53]. The states 'P, E' and 'A/P, P/E' had very low particle numbers (below 500 each, see Supplementary Fig. 1) and resolutions worse than 20 Å. The identification confidence was not high and they were not segmented. Around 3200 particles were in classes that could not be confidently assigned to any one state due to resolution limitations. The 80S*eEF2 class was further subclassified with a shape mask generated from a 12 Å molmap of eIF5a from the PDB model 7OYC[18] to look for heterogeneity in this population (Supplementary Fig. 8).

The first focused classification round with the APE mask was repeated four times with the same settings and the standard deviation for the occurrence frequencies was around 2.5%. Being more cautious

the confidence interval for the reported occurrence frequencies in Fig. 3 might be around 5%.

## Expansion segments model building

To build 2D diagrams of the core of the large subunit rRNA from *D. discoideum*, the incomplete 25S and 5.8S rRNA were downloaded from CRW database[54] and used as initial template. Secondary structure of the conserved part of 5S rRNA was generated in forna package webapp[55]. The initial 2D diagram of the core of the 17S rRNA was generated in R2DT[56] using *D. melanogaster* 18S rRNA as a template. The 2D maps were curated using the modeled 3D structure (described below). The final complete rRNA 2D diagrams were generated by manual incorporation of all the curated structures and the expansion segments. The 3D molecular model of the 80S ribosome of *D. discoideum* was built using available templates and based on density fit. The Sec61-bound Sus scrofa 80S ribosome (PDB: 3J7R[10]), the 80S ribosome of *Drosophila melanogaster* (PDB: 4V6W[8]) and the available part of 25S rRNA from *D. discoideum* (PDB: 5ANB[57]) were used as initial templates. The conserved primary sequences of rRNA were aligned with the respective templates and modeled using ModeRNA web service[58]. The resulting motifs were manually curated in Coot[59], merged, and fitted as rigid body into 80S ribosome map in ChimeraX. The rRNA expansion segments corresponding to the non-aligned part of the sequence were built de novo using RNAComposer[60]. The secondary structures of expansion segments were generated by the forna package webapp and manually incorporated into the final 2D diagram. Alphafold-generated ribosomal protein models were fetched from Uniprot database (https://www.uniprot.org/), aligned to the 80S ribosome templates (PDB: 3J7R[10], 4V6W[8], 5LZS[12]) and fitted into the *D. discoideum* 80S ribosome map. The resulting 80S ribosome model was subjected to several iterations of molecular dynamics flexible fitting in Namdinator web server[61] with the following parameters (map resolution: 8 Å, start temperature: 600 K, final temperature 298 K, G-force scaling factor 0.5, minimization steps 5000, simulation steps 200,000, implicit solvent: exclude, real space refinement cycles: 0). Later, the coordinates of individual rRNA expansion segments were extracted from the 80S ribosome model and their corresponding map fragments were segmented in ChimeraX using Segger[53]. These fragments were additionally processed with Namdinator with default parameters (map resolution: 6 Å start temperature: 298 K, final temperature 298 K, G-force scaling factor 0.3, minimization steps 2000, simulation steps 20,000, implicit solvent: exclude, real space refinement cycles: 5).

## Data visualization and figure preparation

For better visualization of tomographic reconstructions in Fig. 1, bin4 back projected tomograms were deconvolved using Matlab-based scripts (https://github.com/dtegunov/tom_deconv).

For Fig. 2b, and Supplementary Fig. 8, models of the ribosomal protein uL18a and eIF5A (Alphafold-generated models from Uniprot) and a fragment of 26S rRNA (self-built model, see **Expansion segments model building**) were real-space-refined with ISOLDE[62] inside of ChimeraX.

Ribosome translational states were placed back into the corresponding tomographic volume for Supplementary Fig. 4 using the subtomo2ChimeraX scripts (https://github.com/builab/subtomo2Chimera). The distribution of ribosome translational states per tomogram was calculated in Microsoft Excel 2019. Cellular membranes and vesicles were segmented using Amira-Avizo software version 2021.1 (Thermo Scientific) and displayed as isosurface representation. Figures of cryo-EM densities and models were prepared in ChimeraX. Heat maps were generated and FSC curves plotted in GraphPad Prism.

## Reporting summary

Further information on research design is available in the Nature Portfolio Reporting Summary linked to this article.

## Data availability

Cryo-ET density maps generated in this study have been deposited in the EM Data Bank with the following accession codes: EMD-15807 (cytosolic-dataset2-high_res_map), EMD-15808 (cyotosolic-dataset1-high_res_map), EMD-15809 (membrane_bound), EMD-15810 (eEF1A, A/T, P), EMD-15811 (eEF1A, A/T, P, E), EMD-15812 (A, P), EMD-15813 (P), EMD-15814 (factor-bound, A, P), EMD-15815 (eEF2, aa/P, P/E), EMD-15816 (eEF2, ap/P, P/E), EMD-15843 (eEF2-substate1), EMD-15844 (eEF2-substate2) and EMD-15845 (eEF2, P, E).

The previously published structures EMD-4474 (*S. cerevisiae* 80S-Xrn), EMD-5591 (*D. melanogaster* 80 S), EMD-5591 (*M.musculus* 80S-Ebp1), EMD-3069 (*C. lupus* 80S-Sec61-TRAP-OST), EMD-2644 (*S. scrofa* 80S-Sec61) (Supplementary Fig. 5g), and EMD-0202 (*S. cerevisiae* 80S-NatA) are accessible through the Electron Microscopy Data Bank. The previously published ribosome structures 5LZS, 7LS1, 6TNU, 4D61, 4UJC, 4UJD, 3J7R, 4V6W, 5ANB and 7OYC are available through the Protein Data Base. The templates for ribosomal RNA secondary structure diagrams were downloaded from CRW database (https://crw-site.chemistry.gatech.edu/). Source data are provided with this paper.

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

## Acknowledgements

This work was funded by the Max Planck Society and the Chan Zuckerberg Initiative for Visual Proteomics Imaging. P.C.H. is currently supported by an EMBO Fellowship (ALTF 33-2021). We thank Mark Linder from the Central Electron Microscopy facility of the Max Planck Institute of Biophysics for technical support. We thank Özkan Yildiz, Juan F.

Castillo Hernandez, Thomas Hoffmann, Beata Turoňová, Andre Schwarz, Erin Schuman and the Max Planck Computing and Data Facility for support with scientific computing. We thank the Central Electron Microscopy Facility at Max Planck Institute of Biophysics for support with data acquisition.

## Author contributions

M.B. and P.C.H. conceived the project. P.C.H., M.W.T., and S.W. acquired data, P.C.H., J.P.K. and I.K. analyzed data, P.C.H., J.P.K, I.K., M.W.T. and M.B wrote the manuscript. S.W. edited the manuscript. M.B. supervised the project and acquired funding.

## Funding

## Competing interests

The authors declare no competing interests.
