## [Peer Review File · Nature Communications]

Structures of the eukaryotic ribosome and its translational states in situREVIEWER COMMENTS

Reviewer #1 (Remarks to the Author):

In the manuscript entitled “Visualizing translational states of the eukaryotic ribosome in situ” written by Hoffmann et al., authors have solved the structure of 80S ribosomes inside *Dictyostelium discoideum* cells using cryo-electron tomography. Authors have been able to overcome thickness limitations in cryo-electron tomography for larger eukaryotic cells using advancements in cryo-focused ion beam milling. By template matching and extensive 3D classifications, the authors were able to resolve the structure of ribosomes up to 3 Angstrom in their native environment.

The highlights of this manuscript include the classification and identification of different initiation and elongation states of the ribosome inside cells, including tRNA-, eEF1A-, and/or eEF2-bound ribosomes, all of which have been identified in vitro. Additionally, the visualization of the structure of expansion sequences of rRNA is intriguing and it will be useful in determining the role of these rRNA segments in ribosome biogenesis and translation.

I find that this manuscript advances our current knowledge about translational states of ribosomes in their native states in cells. I agree with the vision of the authors that the quantification of the translational states using cryo- electron tomography may become a fingerprint for understanding cellular physiology. Overall, this manuscript provides a useful framework for future studies and is suitable for publication in Nature Communications. However, some of the findings need to be further explored/explained.

1. The authors identified several rRNA expansions in *D. discoideum*. It is unclear how well these expansions are conserved in eukaryotes. A comparison with other eukaryotic rRNA expansions would strengthen the findings.
2. The authors observed the displacement of the ES27L in the membrane-bound ribosomes. It is unclear how the displacement takes place in Fig. S5. Is the ES27L *D. discoideum* specific? A further explanation is warranted.
3. Authors have not been able to identify several other translational states of the ribosomes. Although the structure of 80S-eEF2 is a novel finding, it does not match any known translational states. Is this observation related to the larger variations/heterogeneity in the 80S*eEF2 states observed in the cells?

Reviewer #2 (Remarks to the Author):

The study by Hoffmann et al presents a cryo-ET in situ analysis of ribosomes in *Dictyostelium discoideum* cells. The authors use cutting-edge cryo-FIB/SEM and cryo TEM in conjunction with the highly prolific combination of the software M and RELION to provide snapshots of ribosomes 'in action' at impressive resolution. The data are of high quality, and it is welcome to see such high-resolution 3D images of ribosomes without purification. The local resolution appears to be slightly superior to what has recently been reported for yeast in a preprint (DOI: 10.1101/2022.06.16.496417). The higher resolution (up to 3.0 Å in the core of the LSU compared to 3.5 Å) is likely due to a higher number of particles (~25k), which also allows classifying ribosomes according to their functional states. In summary, 8 states are distinguished. Moreover, *Dictyostelium* appears to have unusually long expansion segments, which can be partially resolved in the in situ ribosomes, differing to some extent between cytosolic and membrane bound particles.

The work is technically on a very high level, which makes it interesting to a larger audience. On the other hand, the molecular interpretation is somewhat superficial and sometimes confusing. The major conclusion of showing the 'potential' to reveal atomic structures is a bit generic and few require convincing of the potential anymore. Thus, it is a bit difficult to discern the major message of the manuscript: is there any specific technological advance that allows getting higher resolution structures? Are there new mechanistic insights that can be derived from the obtained structures? It appears that the expansion segments have the most potential to reveal something unique. In any case, the manuscript requires more thorough discussion in the light of the published ribosome structures – which have implicitly been used for molecular assignments. Detailed comparison to structures of such reconstituted complexes may or may not validate the molecular assignments from the manuscript. In summary, the molecular interpretation appears premature at this point and requires considerable revision.

Major points (in order or occurrence in the text):

- Abstract: the claim of 'atomically resolved' structures does not appear to be justified from the reported findings – atomic resolution is in the realm of 2 Å and in the parts of elongating structures that the paper is about (mainly elongation factors) the resolution presumably barely exceeds 6-7 Å.
- L.30: arguably, the degree of conservation between archaea and eukaryotes is higher than that between bacteria and eukaryotes (a feature typically shared for all cytosolic machinery). Moreover, what does 'most' refer to?

- L. 52: 'structural dynamics' is not an appropriate term for what is observed by cryo-ET (which is static by definition). Cellular cryo-ET images the most abundant states of a molecule. Some aspects of molecular mechanisms may be inferred (not imaged directly), yet short-lived states, which are essential aspects of any 'dynamic' picture, are absent.
- L. 84/85: "... it can potentially interact with chaperones or directly with the nascent chain." Is there any additional support for this hypothesis? The section on expansion segments is most interesting to this reviewer, yet some clearer take-home message would be helpful.
- L. 86: 'membrane-bound': which membrane do the authors refer to? Presumably, only the ER/nuclear envelope (no mitochondria)? If so, are there any notable differences between nuclear envelope and ER?
- L. 87-89: a contact between ER membrane and ribosome via ES27 has been described previously for the mammalian ER (Pfeffer et al, Structure 2012). Does the interaction differ for Dictyostelium?
- L. 92/93: which of the two OST variants (characterized by the catalytic subunits STT3a/b, Ruiz-Canada et al Cell 2009) do the authors refer to? Only the co-translational A variant would be relevant in this case – a quick look at Uniprot suggests that Dictyostelium has only one STT3 gene, which likely codes for the post-translational version. That would explain the absence of OST from the ribosome.
- L. 97/98: the two sentences are confusing. First, 8 states are assigned to elongation – and the following sentence refers to initiation. The referenced figure 3 suggests 6 states in elongation?!
- L. 103: ribosomes without P-site tRNA and with eEF2 bound likely represent 'hibernating' ribosomes (and thus not necessarily 'unexpected'). For some human cells, these were reported to be even the dominant population in the stationary phase. Recent structural studies of the human hibernating ribosomes are, for example, in Brown et al, eLife 2018; Wells et al, PLOS Biol. 2020. That begs the question about more details on the cell culture (the current materials and methods does not provide much detail on fluence), as the amount of hibernating ribosomes is likely a function of cell state.
- L. 112: Fig. S7A may have indeed some potential for further interpretation. Not much is known about the physiological role of hibernating ribosomes at this point and the data clearly show high volatility of this class per individual cell, which may be worth diving into further. Thus, it'd be interesting to further investigate their preferred local habitat in the cell.
- L. 143: there is simply no evidence provided for the assignment of a ribosome class (presumably bottom right in Fig. 3) to decoding (see also general comment on Fig. 3 below).
- L. 142: "In addition, using an in situ approach, we find more than a third of ribosomes engaged at the decoding step, a state which could not be resolved by the ex vivo approach." Which specific states are the authors referring to – and if they have not been seen previously – how can they assign a functional state (decoding)? A number of supplementary figures (and citations) might resolve this enigma.
- L. 144: "On the other hand, the distribution of translational states observed in our dataset is in accord with recent in situ analysis of the translation in prokaryotic Mycoplasma cells". Please specify 'in accord' – there is no reason to believe prokaryotic and eukaryotic translation are identical and the statement appears awkwardly superficial.

- Fig. 3: The nucleotides and identify of elongation factors appear to be inferred from higher resolution structures – which are compared in supplementary figures, not even cited. The primary data provided in the manuscript is insufficient for molecular assignment. For example, the proteome of Dictyostelium exhibits many GTPases that are structurally similar to eEF1a - so how do the authors justify the assignment to eEF1a? The local resolution is likely insufficient for identification. This applies even more so to the assignment of nucleotide states. The main criticism of this reviewer is that there is – from the data and referencing provided in the manuscript – little foundation for the inferred translation cycle.

Minor:

- L. 83: 'unlike its to': remove 'to'

We thank the reviewers for their valuable comments that were very helpful to revise and improve our manuscript. The changes are highlighted in blue and briefly summarized here:

- The section about RNA expansion segments was extended. A structural comparison between different species was included as an additional figure (Supplementary Fig. 5). The main text does now reference the major studies in this area of research.
- More in-depth classification led to sharpened densities for translation factors and tRNAs. The fits of PDB models that lead to the assignment of translation states are now shown in Supplementary Fig. 6 & 7.
- The 80S*eEF2 state was further investigated and subclassified (Supplementary Fig. 8). The two identified substates differ in density at the eIF5A binding site, linking them to previously observed "hibernation"-like ribosome structures in other organisms.

The detailed point by point response follows below:

REVIEWER COMMENTS

Reviewer #1 (Remarks to the Author):

In the manuscript entitled “Visualizing translational states of the eukaryotic ribosome in situ” written by Hoffmann et al., authors have solved the structure of 80S ribosomes inside *Dictyostelium discoideum* cells using cryo-electron tomography. Authors have been able to overcome thickness limitations in cryo-electron tomography for larger eukaryotic cells using advancements in cryo-focused ion beam milling. By template matching and extensive 3D classifications, the authors were able to resolve the structure of ribosomes up to 3 Angstrom in their native environment.

The highlights of this manuscript include the classification and identification of different initiation and elongation states of the ribosome inside cells, including tRNA-, eEF1A-, and/or eEF2-bound ribosomes, all of which have been identified in vitro. Additionally, the visualization of the structure of expansion sequences of rRNA is intriguing and it will be useful in determining the role of these rRNA segments in ribosome biogenesis and translation.

I find that this manuscript advances our current knowledge about translational states of

ribosomes in their native states in cells. I agree with the vision of the authors that the quantification of the translational states using cryo- electron tomography may become a fingerprint for understanding cellular physiology. Overall, this manuscript provides a useful framework for future studies and is suitable for publication in Nature Communications. However, some of the findings need to be further explored/explained.

1. The authors identified several rRNA expansions in *D. discoideum*. It is unclear how well these expansions are conserved in eukaryotes. A comparison with other eukaryotic rRNA expansions would strengthen the findings.

We appreciate the comments of both reviewers regarding rRNA expansion segments. We included a detailed comparison of membrane-bound and cytosolic ribosome structures from *D. discoideum*, baker's yeast, fruit fly and mouse into Supplementary Fig. 5. We discuss the conformation and possible role of ES9S that is rarely found in the other ribosomes and the role of ES7L(D) in ribosome stability in the main text as follows: “*The D. discoideum cytosolic 80S ribosome combines characteristic expansion segment (ES) arrangements of other representative eukaryotic model organisms such as Saccharomyces cerevisiae, Drosophila melanogaster, Mus musculus, Sus scrofa, and Canis lupus (Fig. 2c-e, Supplementary Fig. 5a-h)*⁷⁻¹¹ in a unique manner. The long 17S rRNA ES9S of *D. discoideum* is located at the head of the small subunit and oriented towards the central protuberance of the large subunit where it approaches the long expansion segment ES7L(D) (Fig. 2c, Supplementary Fig. 5a) of the large subunit. Despite the presence of long ES9S in *D. melanogaster* ribosome⁸, and similar three-helical structure of ES7L in *M. musculus*⁹, the intersubunit contact formed by these ES is specific for *D. discoideum* (Supplementary Fig. 5a,c,d). The characteristic 26S rRNA expansion ES27L of *D. discoideum* forms a single helix, and reaches the peptide exit tunnel (Fig. 2c, Supplementary Fig. 5a).”

2. The authors observed the displacement of the ES27L in the membrane-bound ribosomes. It is unclear how the displacement takes place in Fig. S5. Is the ES27L *D. discoideum* specific? A further explanation is warranted.

Our structures do not allow for any conclusions about the exact mechanism of this displacement, e.g. regarding the sequence of events that lead to conformational reorganization. However, in the class of membrane-bound ribosomes the density for ES27L was not visible,

contrasting cytoplasmic ribosomes. The comparison to the relevant structures from other species shown in Supplementary Fig. 5 reveals that ES27L is absent in many of them. Especially when the ribosome is engaged with interaction partners at the peptide exit tunnel or bound to the membrane, their interaction sites tend to overlap with the original position of the ES27L. To make this more transparent, we included the following text into the discussion of the revised manuscript: *“Lastly, the conformation of the ES27L may depend on ribosome localization within the cell. For the D. discoideum 80S cytosolic ribosome (Fig. 2c, Supplementary Fig. 5a), ES27L is protruding towards the exit tunnel of the ribosome, where it could support translation fidelity by recruiting nascent peptide processing enzymes, as was shown for S. cerevisiae (Supplementary Fig. 5h)^{26,27}. In the membrane-bound fraction of the D. discoideum 80S ribosome (Supplementary Fig. 5e), ES27L was only partly resolved, indicating its displacement due to steric overlap with the membrane. Although ES27L was shown to facilitate attachment of the ribosome to microsomes derived from dog pancreas²⁸, we do not observe any density in the corresponding region for D. discoideum (Supplementary Fig. 5e), nor in the other selected ribosomal complexes with Ebp1, or Sec61 (Supplementary Fig. 5d,g). Interestingly, ES27L in the 80S-OST-TRAP complex from C. lupus¹¹ is in a different conformation compared to D. discoideum or Sus scrofa (Supplementary Fig. 5a,e-g). Based on these data we suggest that ES27L is flexible and may adopt various conformations depending on interaction context and cellular localization.”*

3. Authors have not been able to identify several other translational states of the ribosomes. Although the structure of 80S-eEF2 is a novel finding, it does not match any known translational states. Is this observation related to the larger variations/heterogeneity in the 80S*eEF2 states observed in the cells?

Further subclassification of the 80S*eEF2 state revealed two substates which differ in density at the previously described binding site of eIF5A. The density in substate 1 fits eIF5A (Supplementary Fig. 8). Yet, mapping of these substates into the spatial context of the tomogram revealed a pronounced cell to cell variability, in particular for the substate 1 with eIF5A present (Supplementary Fig. 10a), pointing to differences in cellular physiology. We could not identify any specific subcellular localization of the 80S*eEF2 states (exemplified in Supplementary Fig. 10b), nor did we identify morphological differences in the lower magnification overview images of the respective cells in our dataset. We refer to this in the

results part as follows: “*We found that in this dataset translation elongation states are evenly distributed in the perinuclear cytoplasm and comparably populated across different cells (Supplementary Fig. 9a, b). The 80S*eEF2 ribosome substate 1 with density at the eIF5A binding site and the class of unassigned ribosomes showed the largest variation in abundance across cells, indicating some degree of heterogeneity.*”

Reviewer #2 (Remarks to the Author):

The study by Hoffmann et al presents a cryo-ET in situ analysis of ribosomes in Dictyostelium discoideum cells. The authors use cutting-edge cryo-FIB/SEM and cryo TEM in conjunction with the highly prolific combination of the software M and RELION to provide snapshots of ribosomes ‘in action’ at impressive resolution. The data are of high quality, and it is welcome to see such high-resolution 3D images of ribosomes without purification. The local resolution appears to be slightly superior to what has recently been reported for yeast in a preprint (DOI: 10.1101/2022.06.16.496417). The higher resolution (up to 3.0 Å in the core of the LSU compared to 3.5 Å) is likely due to a higher number of particles (~25k), which also allows classifying ribosomes according to their functional states. In summary, 8 states are distinguished. Moreover, Dictyostelium appears to have unusually long expansion segments, which can be partially resolved in the in situ ribosomes, differing to some extent between cytosolic and membrane bound particles.

The work is technically on a very high level, which makes it interesting to a larger audience. On the other hand, the molecular interpretation is somewhat superficial and sometimes confusing. The major conclusion of showing the ‘potential’ to reveal atomic structures is a bit generic and few require convincing of the potential anymore. Thus, it is a bit difficult to discern the major message of the manuscript: is there any specific technological advance that allows getting higher resolution structures? Are there new mechanistic insights that can be derived from the obtained structures? It appears that the expansion segments have the most potential to reveal something unique. In any case, the manuscript requires more thorough discussion in the light of the published ribosome structures – which have implicitly been used for molecular assignments. Detailed comparison to structures of such reconstituted complexes may or may not validate the molecular assignments from the manuscript. In summary, the

molecular interpretation appears premature at this point and requires considerable revision.

We appreciate these comments that were helpful to strengthen the message of our manuscript during the revisions. We respond in more detail below. In very brief, the major novelties of our manuscript are in our view:

- We identify structural states of ribosomes inside of eukaryotic cells. We made an effort to improve our molecular interpretation in this regard. We included two additional supplementary figures (Supplementary Fig. 6 & 7) to make the molecular assignment that was done by fitting of PDB models from reconstituted complexes more transparent. We also expanded discussion of translational states.
- The detailed structural comparison of expansion segments of *D. discoideum* ribosomes to other species adds new insights into the unique arrangement and interaction of expansion segments in *D. discoideum* (please see also our reply to Rev #1, p1 & p2).
- Our paper does not highlight a single technological advance. It rather demonstrates that by combining recent improvements, namely the implementation of more stable energy filters with a very narrow slit, the latest generation of direct electron detectors, better control over thickness and contamination during cryo-lamella preparation and improved subtomogram averaging workflows (WARP/ M refinement and Relion classification), it is possible to obtain unprecedented resolution from eukaryotic cells.

Overall, we believe that our work holds great potential to inspire other researchers to realize *in situ* structural biology projects from cellular lamellae.

Major points (in order of occurrence in the text):

- Abstract: the claim of ‘atomically resolved’ structures does not appear to be justified from the reported findings – atomic resolution is in the realm of 2 Å and in the parts of elongating structures that the paper is about (mainly elongation factors) the resolution presumably barely exceeds 6-7 Å.

We thank the reviewer for this suggestion! We agree and have rephrased the respective part of the abstract to rather highlight the classification of distinct ribosomal subcomplexes in the cellular environment: “*Our work demonstrates the use of in situ structural biology techniques for identifying distinct ribosome states within the cellular environment.*”

- L.30: arguably, the degree of conservation between archaea and eukaryotes is higher than that between bacteria and eukaryotes (a feature typically shared for all cytosolic machinery). Moreover, what does ‘most’ refer to?

We have rephrased this sentence for more clarity as follows: *“Incorporation of every new amino acid into the nascent chain during elongation requires multiple conformational rearrangements of the ribosome and changes to interactions with multiple proteins (translation factors) and RNAs. The factors of the translation cycle involved in elongation are conserved between bacteria and eukaryotes¹.”*

- L. 52: ‘structural dynamics’ is not an appropriate term for what is observed by cryo-ET (which is static by definition). Cellular cryo-ET images the most abundant states of a molecule. Some aspects of molecular mechanisms may be inferred (not imaged directly), yet short-lived states, which are essential aspects of any ‘dynamic’ picture, are absent.

We agree with the reviewer and rephrased this sentence as follows: *“Using 3D classification, we resolved distinct ribosome states bound to translation factors and tRNAs in the cellular environment and assigned them to the translation cycle.”*

- L. 84/85: “... it can potentially interact with chaperones or directly with the nascent chain.” Is there any additional support for this hypothesis? The section on expansion segments is most interesting to this reviewer, yet some clearer take-home message would be helpful.

Previous work suggested that ES27L may contribute to translation fidelity by forming a binding platform for modification enzymes (Fujii et al., Mol cell., 2018; Knorr et al., NSMB, 2019). We nevertheless down-toned this statement because our own work does not provide any direct support for it. Also, in response to Rev.#1 p1 & p2 (please see above), we amended the respective section as follows: *“Lastly, the conformation of the ES27L may depend on ribosome localization within the cell. For the D. discoideum 80S cytosolic ribosome (Fig. 2c, Supplementary Fig. 5a), ES27L is protruding towards the exit tunnel of the ribosome, where it could support translation fidelity by recruiting nascent peptide processing enzymes, as was shown for S. cerevisiae (Supplementary Fig. 5h)^{26,27}. In the membrane-bound fraction of the D. discoideum 80S ribosome (Supplementary Fig. 5e), ES27L was only partly resolved,*

indicating its displacement due to steric overlap with the membrane. Although ES27L was shown to facilitate attachment of the ribosome to microsomes derived from dog pancreas²⁸, we do not observe any density in the corresponding region for D. discoideum (Supplementary Fig. 5e), nor in the other selected ribosomal complexes with Ebp1, or Sec61 (Supplementary Fig. 5d,g). Interestingly, ES27L in the 80S-OST-TRAP complex from C. lupus¹¹ is in a different conformation compared to D. discoideum or Sus scrofa (Supplementary Fig. 5a,e-g). Based on these data we suggest that ES27L is flexible and may adopt various conformations depending on interaction context and cellular localization.”

- L. 86: ‘membrane-bound’: which membrane do the authors refer to? Presumably, only the ER/nuclear envelope (no mitochondria)? If so, are there any notable differences between nuclear envelope and ER?

Indeed the membrane-bound particles are associated with nuclear envelope and the endoplasmic reticulum. Due to the nature of our dataset 1 that was acquired at the perinuclear region of the cells, the largest fraction is bound to the NE. We agree that it would be interesting to look for differences between the NE and ER bound ribosomes, however due to the relatively low number of particles our analysis was not conclusive in that respect. The text was changed accordingly: *"In addition to the cytosolic ribosome, we identified by classification a fraction of ribosomes in dataset 1 with density for associated cellular membrane (3314 particles, Supplementary Fig. 2, Supplementary Fig. 5e). Due to the preferential acquisition of the perinuclear region in dataset 1, a larger fraction of ribosomes was directly bound to the nuclear envelope rather than to the endoplasmic reticulum. 3D classification of these particles could not resolve structural heterogeneity. The membrane-bound ribosome consensus map showed a luminal density proximal to the membrane."*

- L. 87-89: a contact between ER membrane and ribosome via ES27 has been described previously for the mammalian ER (Pfeffer et al, Structure 2012). Does the interaction differ for Dictyostelium?

Yes, the interaction of *D. discoideum* ribosome with the membrane as observed in our study is different from that observed by Pfeffer et al, Structure 2012, who had attributed the membrane connecting density to ES27L. In a later study by Pfeffer et al, Nat Comms 2015 using a similar

sample, the ES27L density was oriented away from the membrane. This further supports our conclusion that ES27L is mobile and may be displaced in a membrane bound ribosome.

- L. 92/93: which of the two OST variants (characterized by the catalytic subunits STT3a/b, Ruiz-Canada et al Cell 2009) do the authors refer to? Only the co-translational A variant would be relevant in this case – a quick look at Uniprot suggests that Dictyostelium has only one STT3 gene, which likely codes for the post-translational version. That would explain the absence of OST from the ribosome.

We thank reviewer for this comment! We agree and changed the discussion section accordingly: *“For the membrane-bound D. discoideum 80S ribosome we observe a density at the luminal side of the membrane, probably attributed to TRAP complex (Supplementary Fig. 5e), but not the glycosylation complex OST, which has been observed in the mammalian 80S-TRAP-OST complex (Supplementary Fig. 5f)¹¹. For mammalian OST, the different catalytic subunits STT3A and STT3B were described to glycosylate either co-translationally on the nascent polypeptide or post-translationally³³. In D. discoideum only one form of the OST subunit STT3 is conserved. The absence of luminal OST in our structure suggests that post-translational N-glycosylation may be the dominant mode for proteins inserted into the nuclear envelope and endoplasmic reticulum membranes in D. discoideum.”*

- L. 97/98: the two sentences are confusing. First, 8 states are assigned to elongation – and the following sentence refers to initiation. The referenced figure 3 suggests 6 states in elongation?!

The text was changed for more clarity. *“This approach revealed nine distinct ribosome structures. With the help of PDB models of ribosomes with associated factors from different species¹²⁻¹⁶, we could assign seven of them to states of the translation-elongation cycle (Fig. 3, Supplementary Fig. 2, Supplementary Fig. 6a-h, Supplementary Fig. 7).”*

The initiation state is now mentioned subsequently in the results part (L122-123): *“The dataset also contained a small fraction of the initiation 80S ribosome (Fig. 3, Supplementary Fig. 6i).”*

- L. 103: ribosomes without P-site tRNA and with eEF2 bound likely represent ‘hibernating’ ribosomes (and thus not necessarily ‘unexpected’). For some human cells, these were reported to be even the dominant population in the stationary phase. Recent structural studies of the

human hibernating ribosomes are, for example, in Brown et al, eLife 2018; Wells et al, PLOS Biol. 2020. That begs the question about more details on the cell culture (the current materials and methods does not provide much detail on fluence), as the amount of hibernating ribosomes is likely a function of cell state.

We thank the reviewer for pointing us to the aforementioned studies and included them into the discussion: *“For several species, different ribosome structures without tRNA ligands, but bound to factors such as eEF2, Habp4, SERBP1, Stm1, Lso2 and eIF5A alone or in combination have been linked to stabilization of dormant ribosomes^{19,36-38}. D. discoideum has genes encoding for a member of the Habp4 protein family and for eIF5A. Our 80S*eEF2 substate 1 map indicates that a binding factor is present between the L1 stalk and the binding site of P-site tRNA, which resembles eIF5A in shape (Supplementary Fig. 8 b). However, further analysis is needed to clarify the cellular role of the eEF2 substates in D. discoideum and if they may progress into functional states of the translation-elongation cycle.”*

We also added more detail to the cell culture methods to make this section more comprehensive to the reader: *“The axenic D. discoideum strain Ax2-214 used in this study carried randomly integrated GFP-Nup62 through a pDEX vector with G418/neomycin resistance cassette 39. Cells were grown in HL5 medium (Formedium) containing 50 µg/mL ampicillin and 20 µg/mL geneticin G418 (Sigma Aldrich) at 20 ± 2 °C. Cells were kept either as adherent cells in subconfluent conditions or as suspension culture at a cell density between 1 x10⁵ cells/ml to 4 x10⁶ cells/ml. Cells were sub-cultured for a maximum of four weeks before re-growing them from cryo-stocks. Before grid preparation, cells were adjusted to a concentration of 2-3 x10⁵ cells/ml and allowed to adhere for 2-4 hrs prior to cryo-fixation.”*

- L. 112: Fig. S7A may have indeed some potential for further interpretation. Not much is known about the physiological role of hibernating ribosomes at this point and the data clearly show high volatility of this class per individual cell, which may be worth diving into further. Thus, it'd be interesting to further investigate their preferred local habitat in the cell.

Indeed, we were also interested in the high volatility of this class and were thankful for this suggestion by the reviewer. Further subclassification of the 80S*eEF2 state revealed two

substates which differ in their density at the binding side of eIF5A (Supplementary Fig. 8). The mapping of these states into the spatial context of the tomogram revealed a pronounced cell to cell variability, in particular for the substate with factor density (Supplementary Fig. 10a), pointing to differences in their physiology. However, we could not identify any specific subcellular localization of the 80S*eEF2 states (exemplified in Supplementary Fig. 9b), nor did we identify morphological differences from the lower magnification overview images of the respective cells. We refer to this in the results part as follows: *“We found that in this dataset translation elongation states are evenly distributed in the perinuclear cytoplasm and comparably populated across different cells (Supplementary Fig. 9a, b). The 80S*eEF2 ribosome substate 1 with density at the eIF5A binding site and the class of unassigned ribosomes showed the largest variation in abundance across cells, indicating some degree of heterogeneity.”*

- L. 143: there is simply no evidence provided for the assignment of a ribosome class (presumably bottom right in Fig. 3) to decoding (see also general comment on Fig. 3 below).
- L. 142: “In addition, using an in situ approach, we find more than a third of ribosomes engaged at the decoding step, a state which could not be resolved by the ex vivo approach.” Which specific states are the authors referring to – and if they have not been seen previously – how can they assign a functional state (decoding)? A number of supplementary figures (and citations) might resolve this enigma.

We agree with the reviewer, that the respective states cannot clearly be assigned to process of decoding and edited the respective text as follows: *“Using the in situ approach, we find more than a third of ribosomes engaged at the pre-peptide bond formation steps, states which could not be resolved by the ex vivo approach. We could not observe all states that had been identified ex vivo, possibly due to low abundance of the respective particles. Therefore, both ex vivo and in situ studies have their benefits and result in a complementary view of the conformational landscape of ribosomes.”*

In order to show how ribosome states were assigned to translational steps, we have added two additional supplementary figures (Supplementary Fig. 6 and Supplementary Fig. 7) to show how we assigned the obtained ribosome maps to certain states with the help of PDB models of 80S ribosomes with factors and tRNAs bound, which are now also appropriately referenced throughout the manuscript.

- L. 144: “On the other hand, the distribution of translational states observed in our dataset is in accord with recent *in situ* analysis of the translation in prokaryotic *Mycoplasma* cells”. Please specify ‘in accord’ – there is no reason to believe prokaryotic and eukaryotic translation are identical and the statement appears awkwardly superficial.

We thank the reviewer for pointing this out and agree that the phrasing of this sentence was misleading. We have expanded the respective section in the discussion: “*A recent study of in situ translation in prokaryotic Mycoplasma cells observed the largest fractions of ribosomes in the peptidyl-transfer step and the EF-Tu bound pre-peptide bond formation steps*³. Thus, prokaryotic *Mycoplasma* cells and eukaryotic *D. discoideum* cells show notable similarity in their *in vivo* distribution of translation-elongation states.”

We believe it is important to compare both studies as they represent the only *in vivo* studies up to date and both organisms show notable similarities in distributions of ribosome states of the translation-elongation cycle. If that will persist in further studies to come, remains to be seen.

- Fig. 3: The nucleotides and identify of elongation factors appear to be inferred from higher resolution structures – which are compared in supplementary figures, not even cited. The primary data provided in the manuscript is insufficient for molecular assignment. For example, the proteome of *Dictyostelium* exhibits many GTPases that are structurally similar to eEF1a – so how do the authors justify the assignment to eEF1a? The local resolution is likely insufficient for identification. This applies even more so to the assignment of nucleotide states. The main criticism of this reviewer is that there is – from the data and referencing provided in the manuscript – little foundation for the inferred translation cycle.

We did not intend to make any claims about the nucleotide status of translation factors. The GTP/GDP states that were indicated in the previous version of Figure 3 were inferred from literature but not our data. We have removed this from the respective scheme in the revised version.

We have revised the manuscript and included the appropriate citations for all PDB models used for assigning our cryo-ET maps to specific ribosomes states. Indeed, the resolution of the maps is not sufficient for molecular assignment based on their atomic structure. For the apparent resolution range, we rely on fitting previous structures into our *in situ* maps. In the revised version, we show fits of the assigned elongation factors and tRNAs with the corresponding

segmented density in Supplementary Fig. 6. We furthermore show how other potential GTPases fit into the density for the most abundant eEF1A*tRNA state in comparison to eEF1A*tRNA (Supplementary Fig. 7a), the other small GTPases do not fit well into the density then the respective density (Supplementary Fig. 7b-e).

Minor:

- L. 83: 'unlike its to': remove 'to'

This has been edited as suggested.

REVIEWERS' COMMENTS

Reviewer #1 (Remarks to the Author):

The authors have addressed the comments nicely.

Reviewer #2 (Remarks to the Author):

The authors addressed most of the points raised in their revision and the manuscript improved substantially. A major change of the manuscript is, however, that the authors reclassified part of their data, resulting in a notably different elongation cycle. While the authors were confident to assign 3 eEF1a-bound, 2 eEF2-bound and 1 factor-less states in the previous version of the manuscript, the authors now assign their data to 2 eEF1a, 4 eEF2 and 1 state without elongation factors in the elongation cycle.

This change of assignments highlights a remaining issue with the manuscript, which refers to Figure 3. The authors point out that they want to demonstrate 'the potential' of cryo-ET for classifying distinct translational states, which the reviewer accepts. Nevertheless, as it is presented now Figure 3 creates the impression that the study goes beyond showing the potential, rather identifying new states of the elongation cycle already. The changes during the revision suggest that molecular assignments of low-abundance states appear to be rather tentative than definite. This is not a problem per se but should be reflected in the language.

While most elongation states seem to be backed by previous higher resolution data (which should be cited appropriately, see below), the 'eEF2-compact state' and its placement in the elongation cycle appear highly speculative at the resolution obtained. This reviewer would strongly advise more caution in the assignment of elongation factors at this level of resolution. The recommendation would be to claim 'assignment' only for those intermediates that have support from in vitro studies (i.e., ribosome structures in complex with the respective elongation factors), while clarifying the speculative or tentative character of other states. In summary, the study shows exciting data, but the authors should not feel pressured to drawing too venturous conclusions from low-resolution data they may regret later (or clearly mark them as hypotheses). Textual revisions and consideration of below suggestions to improve Figure should make the manuscript acceptable for publication in Nature Communications.

Specific points:

- The various eEF2 states are reconstructed from a modest 700 - 1,600 particles, resulting at global resolutions mostly exceeding 10 Å and notably worse at the elongation factor site. At this resolution the unambiguous identification of proteins among similar folds appears barely possible. The authors even go one step further and decompose eEF2 into its domains and fit those independently ('EF2-compact'). This reviewer does not consider the resolution sufficient for unambiguous identification and domain fitting. Comparison to structures from biochemical reconstitution may come to help here to make assignments plausible, as done in the recent work by Xue et al (Nature 2022). Here, the assignments of eEF2/EF-G were supported by single particle work from the Fischer and Rodnina labs (Petrychenko, Nat. Comms. 2021). Nevertheless, there seems to not exist orthogonal data for the EF2-compact state and the Supplementary Fig. 6 is also not supportive of a compelling fit. The reviewer's recommendation would be to refrain from an assignment of the binding factor(s) and positioning in the elongation cycle. If the authors wish to speculate on the assignment of eEF2, it should be indicated as such.
- The SSU rotation states are neither indicated in Fig. 3 nor in Fig. S2, which makes it difficult to follow the cycle and the plausibility of the assignments. For example, this is nicely visualized using different colors in the recent in situ visualization of the bacterial elongation cycle (Xue et al, Nature 2022). Alternatively, the rotation states may simply be added as text.
- In line with the suggestion to compare intermediate states to those published this reviewer also suggests complying with the common nomenclature for eEF2/EF-G. Again, Xue et al may serve as a reference. The terms A/P/P,P/E and A/A/P,P/E appear uncommon.
- The P-only state is assigned to initiation. Why not as part of the elongation cycle as in (Xue et al, Nature 2022)? Thus, the state would be an alternative path to 'eEF1a A/T, P'. The occurrence of this state should be more frequent during elongation of a peptide than initiation.

Minor

- L. 203/204: The authors compare Dictyostelium translation states to human ones. While the preparation involved in the ex vivo study may indeed have major influence on abundance of elongation factors, at this point species differences cannot be neglected.

We thank reviewer #2 for the valuable comments. We now included the suggested revisions as explained in the detailed point-by-point response below and highlighted the altered text passages in blue.

REVIEWERS' COMMENTS

Reviewer #1 (Remarks to the Author):

The authors have addressed the comments nicely.

Reviewer #2 (Remarks to the Author):

The authors addressed most of the points raised in their revision and the manuscript improved substantially. A major change of the manuscript is, however, that the authors reclassified part of their data, resulting in a notably different elongation cycle. While the authors were confident to assign 3 eEF1a-bound, 2 eEF2-bound and 1 factor-less states in the previous version of the manuscript, the authors now assign their data to 2 eEF1a, 4 eEF2 and 1 state without elongation factors in the elongation cycle.

This change of assignments highlights a remaining issue with the manuscript, which refers to Figure 3. The authors point out that they want to demonstrate ‘the potential’ of cryo-ET for classifying distinct translational states, which the reviewer accepts. Nevertheless, as it is presented now Figure 3 creates the impression that the study goes beyond showing the potential, rather identifying new states of the elongation cycle already. The changes during the revision suggest that molecular assignments of low-abundance states appear to be rather tentative than definite. This is not a problem per se but should be reflected in the language.

We agree with the reviewer that the classification of states shown in Figure 3 should not be perceived as (near-)definitive picture of eukaryotic ribosome translation. Although we do mention in the manuscript that these structural approaches are most likely influenced by particle numbers and data quality, and that there is cell-to-cell variability and possibly differences between species, we agree that resolving intermediate states with low abundance is challenging and the depiction of ribosome state distribution should reflect a more global

view on translation. We therefore show only rounded numbers and refrain from showing percentages in Figure 3.

While most elongation states seem to be backed by previous higher resolution data (which should be cited appropriately, see below), the 'eEF2-compact state' and its placement in the elongation cycle appear highly speculative at the resolution obtained. This reviewer would strongly advise more caution in the assignment of elongation factors at this level of resolution. The recommendation would be to claim 'assignment' only for those intermediates that have support from in vitro studies (i.e., ribosome structures in complex with the respective elongation factors), while clarifying the speculative or tentative character of other states.

Following the very valid reviewer suggestion, we have now removed the eEF2-compact state from the elongation cycle, renamed 'eEF2-compact' as 'factor-bound' state, and show it together with the other two states that could not be placed into the translation cycle in a separate subpanel of Figure 3b. We have changed the text and figure legend accordingly and explicitly mention the speculative nature of the 'putative factors'.

We have also added the appropriate citations to all elongation states which are backed by higher resolution data.

In summary, the study shows exciting data, but the authors should not feel pressured to drawing too venturous conclusions from low-resolution data they may regret later (or clearly mark them as hypotheses). Textual revisions and consideration of below suggestions to improve Figure should make the manuscript acceptable for publication in Nature Communications.

Specific points:

- The various eEF2 states are reconstructed from a modest 700 - 1,600 particles, resulting at global resolutions mostly exceeding 10 Å and notably worse at the elongation factor site. At this resolution the unambiguous identification of proteins among similar folds appears barely possible. The authors even go one step further and decompose eEF2 into its domains and fit those independently ('EF2-compact'). This reviewer does not consider the resolution

sufficient for unambiguous identification and domain fitting. Comparison to structures from biochemical reconstitution may come to help here to make assignments plausible, as done in the recent work by Xue et al (Nature 2022). Here, the assignments of eEF2/EF-G were supported by single particle work from the Fischer and Rodnina labs (Petrychenko, Nat. Comms. 2021). Nevertheless, there seems to does not seem to exist orthogonal data for the EF2-compact state and the Supplementary Fig. 6 is also not supportive of a compelling fit. The reviewer's recommendation would be to refrain from an assignment of the binding factor(s) and positioning in the elongation cycle. If the authors wish to speculate on the assignment of eEF2, it should be indicated as such.

We have followed the reviewer's suggestion and refer to the comparable states seen by Xue et al. (Nature 2022) and Petrychenko et al. (Nat. Comms. 2021) in the text and added the appropriate citations:

"Two of these states show different conformations of eEF2 and A/P- and P/E-tRNAs (Fig. 3a, Supplementary Fig. 6c, d) comparable to prokaryotic translational states with EF-G^{3,17}"

In addition, we have removed the eEF2 compact state from the translation cycle in Figure 3 and instead show it together with the other states that could not be placed in the translation cycle in a new panel in Figure 3b. We also explicitly mention the tentative nature of the factors in this class in the discussion part of the manuscript and renamed eEF2 compact state to 'factor-bound' accordingly. Results and Discussion parts were changes to the following:

" In addition, we observed three factor-bound states, which we could not directly place in the translation cycle (Fig. 3b). This included two eEF2 bound 80S states without tRNA that differed in density at the previously described binding site for eIF5A¹⁸⁻²⁰ (Fig. 3b, Supplementary Fig. 8a, b). One other state contained A/A and P/P-tRNA, and an additional density at the translation factor binding site, but could not be immediately attributed to eEF1A or eFE2 in its elongated form (Fig. 3b, Supplementary Fig. 8c). "

"Besides the well described states of the eukaryotic translation-elongation cycle, we identify a state with a compact density at the translation factor binding site and A/A and P/P-tRNA. Due to lack of atomic models and the limited local resolution of the factor density, it is not possible

to confidently determine its identity. However, fitting of the individual domains of eEF2 into the density may suggest a compact conformation of eEF2, which had been described to exist for the bacterial factor EF-G^{35,36}."

- The SSU rotation states are neither indicated in Fig. 3 nor in Fig. S2, which makes it difficult to follow the cycle and the plausibility of the assignments. For example, this is nicely visualized using different colors in the recent in situ visualization of the bacterial elongation cycle (Xue et al, Nature 2022). Alternatively, the rotation states may simply be added as text.

We thank the reviewer for this suggestion; to make it easier to follow the translation-elongation cycle we have added the rotation states in writing to Figure 3.

- In line with the suggestion to compare intermediate states to those published this reviewer also suggests complying with the common nomenclature for eEF2/EF-G. Again, Xue et al may serve as a reference. The terms A/P/P,P/E and A/A/P,P/E appear uncommon.

We have changed the naming of these states to aa/P, P/E and ap/P, P/E to adopt common terms for the intermediate states and explain the naming in the figure legend.

- The P-only state is assigned to initiation. Why not as part of the elongation cycle as in (Xue et al, Nature 2022)? Thus, the state would be an alternative path to 'eEF1a A/T, P'. The occurrence of this state should be more frequent during elongation of a peptide than initiation.

This alternative was discussed by the authors - however because the particle number for this state is limited, we cannot make any claims about the presence of nascent chain in this class, which we would expect in a scenario where this serves as alternative path to eEF1A A/T, P during elongation. For this reason, we decided to conservatively assign the P-only state to initiation.

Minor

- L. 203/204: The authors compare Dictyostelium translation states to human ones. While the

preparation involved in the ex vivo study may indeed have major influence on abundance of elongation factors, at this point species differences cannot be neglected.

Following the reviewer's advice, we have added a sentence about the possibility of species-specific differences of translation states in line 198:

"Despite the similarity between these two organisms, possible differences of translation states between other species should be considered."